# Exploring the spatiotemporal dynamics and resilience assessment of urban networks from the perspective of population flow

Jiulin Li[1,2]*, Wenhui Lin[1,2], Jinlong Chu[1,2]

**1** School of Architecture and Planning, Anhui Jianzhu University, Hefei, China, **2** Anhui Collaborative Innovation Center for Urbanization Construction, Hefei, China

* ljiul90@163.com

## Abstract

The spatial patterns of population mobility serve as a critical indicator for urban network characterization, providing an essential foundation for resilience assessment. Based on the complex network theory, this study constructs urban networks using the Baidu Migration Big Data of the Jiangsu-Zhejiang-Shanghai region in 2023 to analyze the spatiotemporal dynamics of population flow. Through a research framework of "structure measurement-scenario simulation-resilience assessment," the study systematically reveals the response mechanisms of urban networks. The static characteristics of network resilience in the normal scenario and the dynamic characteristics in the disruption scenario were analyzed. The results are as follows: (1) Population flow is dense in the central region and sparse in the north and south. Network clusters exhibit dual characteristics of "administrative boundary constraints" and "economic gravity dominance", forming more easily among developed cities across provinces or adjacent cities within the same province. The overall connection intensity of the network during holidays is markedly higher than that in the daily period. However, daily contact between developed cities is more frequent than that during holidays, indicating strong intercity commuting and routine movement. (2) In the normal scenario, core cities possess prominent centrality, while the hierarchy of the network is less pronounced. The agglomeration among nodes is moderate but features evident asymmetric connections. The transmission efficiency is relatively high. (3) In the disruption scenario, both the network transmission efficiency and the path connectivity experience phased changes, and the impact of deliberate disturbances on resilience is more significant than that of random disturbances. A handful of cities with crucial influence constitute the core network. This research aims to reveal the resilience characteristics and response mechanisms of population flow networks, offering insights into regional spatial coordination and sustainable development.

**Data availability statement:** The dataset is available in Mendeley Data (https://data.mendeley.com/) with the DOI: Lin, Wenhui (2025), "Data collected from the Baidu Migration Platform", Mendeley Data, V1, doi: 10.17632/5zp632d7wm.1.

**Funding:** This research was funded by the Anhui Office of Philosophy and Social Science, grant number AHSKD2023D028. The grant recipient is Jiulin Li, and his ORCID number is 0009-0001-1967-8124. He was responsible for conceptualization, funding acquisition, methodology, resources and writing – review & editing.

**Competing interests:** The authors have declared that no competing interests exist.

## 1. Introduction

As economic globalization continues to drive the development of regional spatial patterns, the interdependence and influence of cities have reached unprecedented levels. Various studies on urban development are no longer limited to the individual city itself, but focus more on its role and positioning in the region [1]. Therefore, as a form of network-like spatial organization, the urban network has attracted much attention from the academic community. As the inter-city connections become more extensive and dense, the challenges and risks that urban networks face become more diversified. Enhancing the resilience of urban networks and improving the self-healing ability of cities to cope with risks has become an essential issue in urban development research [2–4]. At present, scholars have constructed urban spatial correlation networks from the perspectives of industrial economy [5,6], transportation [7,8], innovation and technology [9–11], tourism [12–14], etc., and used gravitational models, neural network prediction models, geographic probes, Bayesian network to investigate the location and structural characteristics, evolutionary patterns and influencing factors of urban nodes in the network, which indicates that the structure of the urban network is highly correlated with the level of regional resilience. Many studies [15,16] focus on the static characterization of urban network resilience, while there is a lack of in-depth research on the quantitative assessment of network resilience changes caused by local node failures. Actually, attacks on different nodes may affect the network resilience level to various degrees, and the resulting cascading effect may further constrain regional security and stability [17–19]. Therefore, theoretically and practically, a comprehensive analysis of the network resilience characteristics under natural and disruption scenarios and its response mechanism in the face of shocks is highly significant.

Population flow is the primary carrier of various flow elements such as socioeconomic systems, urban transportation, regional culture, etc. [20,21]. Under the trend of increasingly intertwined flow elements and physical space, population flow can indicate the scale of elements carried by nodes in the urban network and the strength of the connection between nodes, thus becoming a critical attribute for measuring the resilience of the network structure. The rapid development of urbanization in China is closely related to the geographic behavior of large-scale interregional population flow. According to the seventh China Population Census, China's mobile population is about 367 million people, accounting for 26.0% of the total population, which is the decade with the largest increase in the number of mobile population since the census. While the average annual population growth rate was only 0.51%, the average yearly growth rate of the mobile population was as high as 6.97%. Therefore, scientific analysis and rational management of the regional mobile population are an integral part of realizing synergetic regional development in a certain period in the future [22,23]. The rapid development of information technology and the Internet has facilitated the accurate acquisition and collection of various kinds of objective, dynamic, and diversified geospatial and spatial big data [24,25], which provides reliable support for urban network research [26,27]. Urban migration networks based

on geospatial "mobility data" have received increasing attention. Studies have been conducted to analyze the characteristics and causes of population flow during specific events (e.g., COVID-19 [28–30], specific periods (e.g., Spring Festival [31,32], Mid-Autumn-National Day Holidays [33,34], and ordinary weekdays [35,36]), and specific areas (e.g., a specific city [37], a province [38], an urban agglomeration [39], or even a country [40]), which have greatly enriched the perspective of population flow research. Many studies have only focused on analyzing the structural characteristics of population flow networks in a specific period [41,42]. They thus cannot comprehensively and objectively characterize the changes in the characteristics and resilience level of population flow networks in cities in different periods. This study takes the Jiangsu-Zhejiang-Shanghai region, where the intensity of population flow is relatively high, as the research object. The population flow network is used to characterize the urban network. By employing the complex network theory as an analytical tool, we conduct an in-depth exploration of the characteristics intrinsic to urban networks and disclose the spatiotemporal patterns manifested by the population flow across diverse periods. The static characteristics of network resilience within the normal scenario and its dynamic characteristics under the disruption scenario simulation are appraised to suggest spatial optimization approaches. This research offers a scientific guideline for enhancing the robustness of regional urban networks and optimizing the competitive associations among cities.

## 2. Materials and methods

### 2.1 Study area

The Jiangsu-Zhejiang-Shanghai region is located in eastern China downstream of the Yangtze River. As of 2023, the region contains 25 cities (Fig 1), with a land area of about 219,000 km2 and a resident population of about 16,101.4 million people. Compared to other regions, it has implemented regional integration strategies earlier and has a higher level of regional integration and development, making it one of the most influential and promising regions in China and globally. The elevated population flow density has substantially influenced the evolution of the regional spatial pattern. According to the data from the Seventh National Population Census of China, the floating populations in Jiangsu Province, Zhejiang Province, and Shanghai all exceed 10 million. Notably, the mobile population in Shanghai constitutes over 42% of the total population, ranking first among all provinces (including municipalities directly under the central government) in China. Zhejiang Province and Jiangsu Province follow in second and third place, respectively, in terms of the size of their mobile populations. Consequently, an investigation into the attributes of urban network structure and resilience in this region, from the vantage point of population flow, is eminently representative and offers significant insights for network resilience research in other regions.

### 2.2 Source of data

The data originate from the Baidu Migration Platform (https://qianxi.baidu.com), which draws upon the analytics platform LBS (Location Based Services) to compile the daily migration scale index of each city, as well as the ratio of the scale of inbound and outbound migration between cities. This enables the visualization of the strength of the linkage of population flow and the direction of the trajectory. This study employs approximately 440,000 migration records from 25 cities in the Jiangsu-Zhejiang-Shanghai Region, spanning the period from January 1, 2023, to December 31, 2023 (365 days). These data are utilized to construct population flow networks, thereby enabling the characterization of the city network.

### 2.3 Research framework

Complex network theory is valuable for investigating the multivariate relationships between network nodes. It offers a quantitative analysis of network structure, providing insights into the research process and becoming an essential component in developing a framework for measuring urban network resilience [43,44]. Firstly, the urban network is constructed using data on the scale and direction of inter-city population flow, and the characteristics of the network connections in

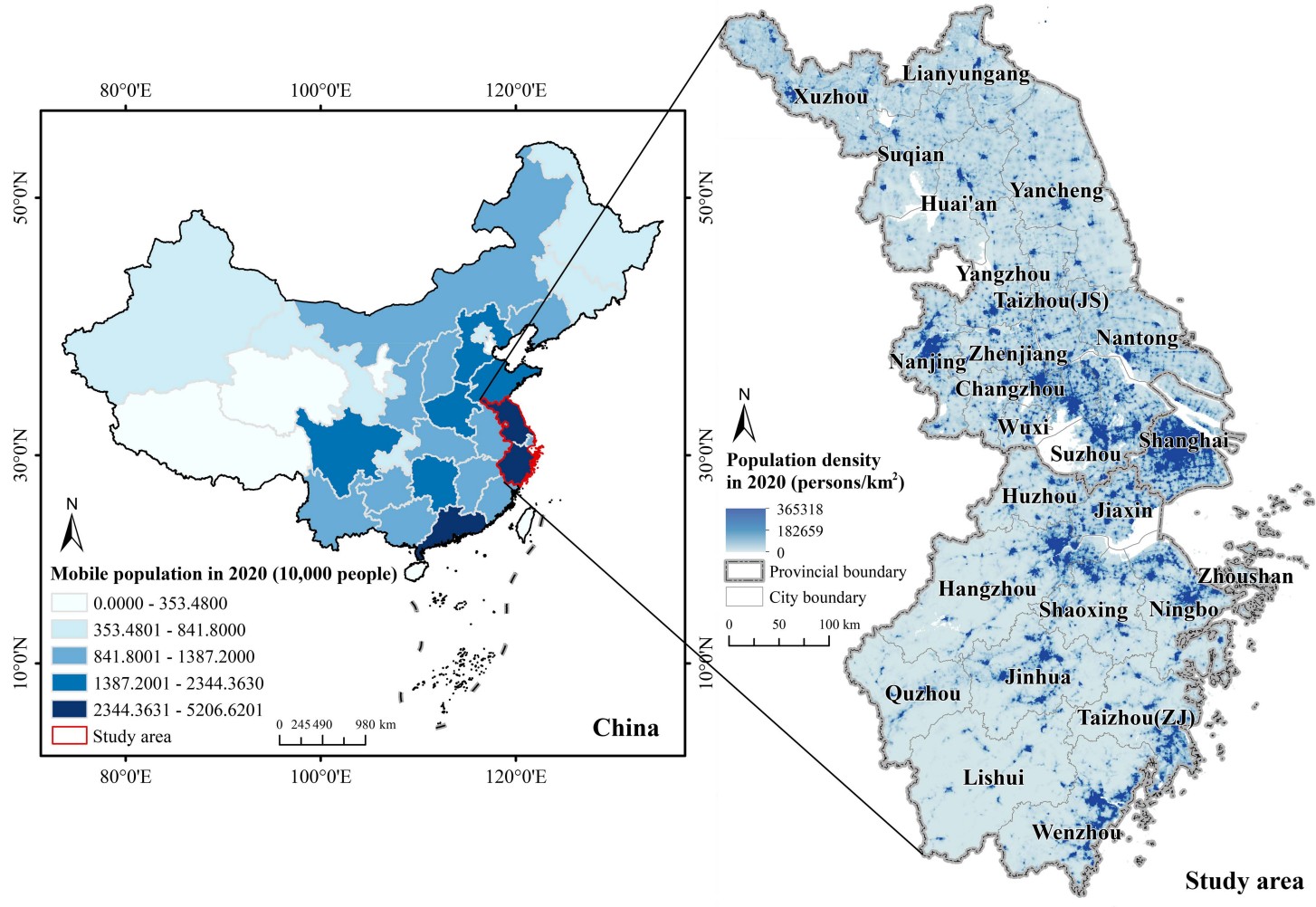

**Fig 1. Location map of the study area.** Reprinted from [(http://www.resdc.cn/DOI),2023.DOI:10.12078/2023010101] under a CC BY license, with permission from [Xu Xinliang], original copyright [January 2023].

different periods are analyzed. Secondly, the complex network theory is employed to investigate the resilience of urban network structures in typical circumstances from the perspective of resilience. Ultimately, with the assistance of a computer program to simulate disruption scenario, the network nodes and path failures are simulated by random and deliberate attacks. The network resilience change eigenvalues are then evaluated, and the core network, the relatively intact network, and the edge nodes are identified.

Referring to related studies [45,46], the current assessment indicators of network structural resilience have not yet formed a unified standard but emphasize the selection of indicators with a focus on the characteristics of regional development. Considering the characteristics of high population density and high urbanization rate in the study area, the main factors affecting the network structural resilience are combined with the coordinated development of the region and the benign competition between cities to select the indicators (Fig 2). By comprehensively using Ucinet software to analyze centrality, hierarchy, agglomeration, and transmission, this tool measures the level of network structural resilience under normal scenario from seven indicators: degree centrality, betweenness centrality, closeness centrality, composite

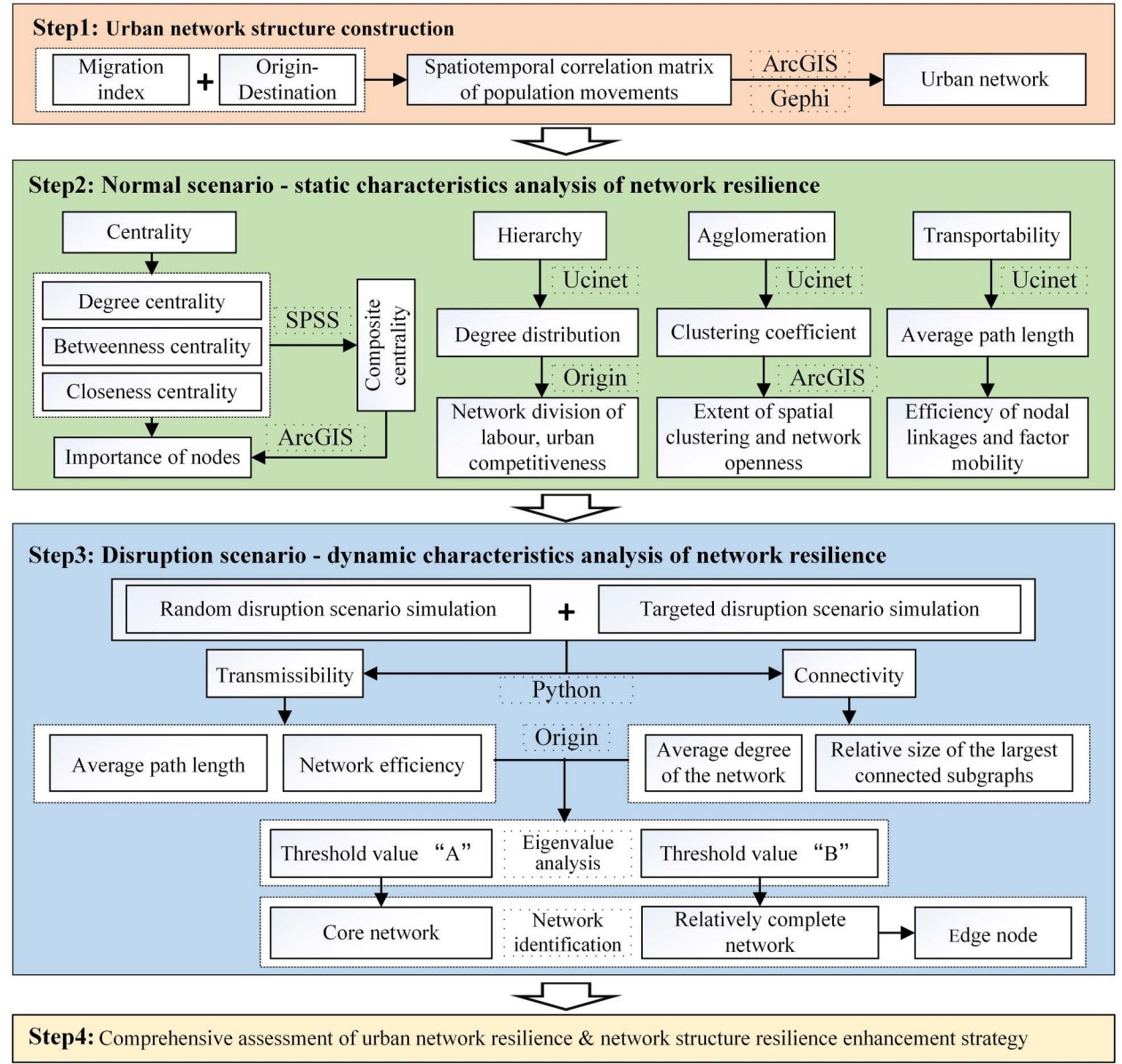

**Fig 2. Research framework of urban network resilience.**

centrality, degree distribution, agglomeration coefficient, and average path length [47,48]. Transmission and connectivity are analyzed using the Network X network analysis tool, which measures the change in network structural resilience under interrupted simulation from four metrics: average path length, network efficiency, average degree of the network, and relative size of the maximum connectivity subgraph. The Origin data analysis utility is employed to distinguish the central and peripheral nodes in the network architecture. These metrics comprehensively reflect the network characteristics and structural robustness from various dimensions, which are important benchmarks for optimizing the network structure and improving the network robustness. Based on the above methods, a research framework of urban network resilience is constructed, which integrates "structure analysis-scenario simulation-resilience assessment".

## 2.4 Methods

### 2.4.1 Method of network construction.

According to the complex network theory, cities are considered as nodes, and population migration interactions between cities are modelled as weighted edges [49]. Leveraging daily datasets from the Baidu Migration Platform—including daily total migration index ($T$), origin-destination pairs, and directional migration proportions ($I$) — we constructed 365 daily 25×25 connection matrices to represent the population flow network among 25 cities over a year. The average daily migration scale index, $T_{day}$ is calculated by averaging the bidirectional connection strength between cities over a specified period, reflecting the intensity of population flow interactions. Higher $T_{day}$ values signify intensified population mobility interdependence between city pairs. The formula is:

$$T_{day\,(i,j)} = \frac{1}{D} \sum_{d=1}^{D} (T_i \times I_{ij} + T_j \times I_{ji})$$

(1)

where D denotes the number of days in a given period. $T_i$ and $T_j$ are the migration indices of cities i and j, respectively. $I_{ij}$ denotes the proportion of migration from city i to city j, and $I_{ji}$ denotes the proportion of migration from city j to city i.

### 2.4.2 Methods of scenario simulation.

(1) Normal scenario: static characteristics analysis of network resilience

The normal scenario represents the undisturbed state of the urban network, where all nodes and edges operate under routine conditions, reflecting the inherent structural resilience of the system. Four network attributes-centrality, hierarchy, agglomeration, and transmissibility-were analyzed to assess the global significance of nodes [8], evaluate the hierarchical organization and redundancy of the division of labor, quantify the cohesion of subgroups and measure the overall flow efficiency of the network. This multidimensional analysis reveals the static characteristics of the network in the normal scenario, i.e., the functional structural characteristics of the urban network and its operation mode, which provides a strategic reference for optimizing the regional development configuration.

(2) Disruption scenario: dynamic characteristics analysis of network resilience.

Targeted disruption and random disruption are two fundamental methods for studying resilience changes in complex networks under disruption scenario [50]. Firstly, targeted disruption involves intentional interventions to alter urban network structure or function, typically driven by human factors aiming to modify system operations or achieve specific impacts. Using Network X, nodes are removed sequentially based on their composite centrality rankings (from highest to lowest), simulating extreme adverse changes that the network may encounter. Secondly, random disruption refers to introducing stochastic changes or disturbances into the urban network structure to simulate systemic uncertainties. Such disruptions may originate from external factors (e.g., natural catastrophes, black swan events) or internal random variations, both of which can unpredictably impact network structure and functionality. Using the complex network analysis tool Network X, nodes and edges are randomly removed one by one to simulate future uncertainties. Finally, by comparing metric changes in network transmissibility and connectivity under targeted disruption versus random disruption, the study quantifies disruption impacts on resilience. Key vulnerabilities (nodes causing significant resilience decline) and critical resilience thresholds are identified, informing region-specific optimization strategies.

### 2.4.3 Methods of measuring the network structural resilience.

(1) Centrality: degree centrality, betweenness centrality, closeness centrality, composite centrality

Centrality is used to evaluate the importance or influence of a node in a network. The degree indicates the number of edges that are directly connected to a node. Degree centrality, $D_i$, can measure how closely a node is connected to other nodes in the network [51]; the higher its value, the more dominant the node is in the network. It is calculated as follows:

$$D_i = \frac{\sum_{j=1}^{N} D_{ij}}{(N-1)} \tag{2}$$

where $N$ is the number of nodes in the network, $D_{ij}$ value is 1 when node j is connected to $i$; otherwise, $D_{ij}$ is 0.

The betweenness centrality, $B_i$, is a measure of a node's transit and bridging ability through the number of shortest paths of a node in the network [51]. The higher its value, the more influential the node is in resource scheduling and information dissemination. The formula is as follows:

$$B_i = \sum_{i \neq j} \frac{N_{ij}(q)}{N_{ij}} \tag{3}$$

where $N_{ij}$ is the number of shortest paths between nodes $i$ and $j$, and $N_{ij}(q)$ is the number of shortest paths between nodes $i$ and $j$ passing through node $q$.

Closeness centrality $A_i$ denotes the sum of the shortest distances from the node to all other nodes in the network [52]. The smaller its value, the more important the node is in the network. The formula is as follows:

$$A_i = \frac{(N-1)}{\sum_{j=1, i \neq j}^{N} L_{ij}} \tag{4}$$

where $N$ is the number of nodes in the network, and $L_{ij}$ is the shortest distance between nodes $i$ and $j$.

Composite centrality $Z_i$ comprehensively reflects the breadth of connectivity, hub control capacity, and information transmission efficiency of nodes, enabling a thorough analysis of node importance. To assess node significance multidimensionally, composite centrality is calculated based on degree centrality, betweenness centrality and closeness centrality [50]. To eliminate dimensional discrepancies, z-score normalization was implemented. The entropy weight method in SPSS was employed to calculate the weight coefficients of the three centralities, resulting in the composite centrality for each node. The formula is:

$$Z_i = \alpha \times D_i* + \beta \times B_i* + \gamma \times A_i* \tag{5}$$

Where α, β, γ are the weights assigned to standardized degree centrality $D_i$*, betweenness centrality $B_i*$, and closeness centrality $A_i*$, respectively.

(2) Hierarchy: degree distribution.

The degree distribution represents the frequency or probability profile of the degrees of all the nodes in the network. With the rank-size rule, the degree distribution of the network is obtained by linearly fitting the degree values of the nodes in order from largest to smallest [53]. The larger the slope, the more significant the network hierarchy. The larger its slope, the more significant the network hierarchy. The formula is as follows:

$$K_i = D \times (K_i^*)^a \tag{6}$$

The treatment of Eq:

$$ln(K_i) = \ln(D) + \ln(K_i^*) \tag{7}$$

where $K_i$ is the degree of node $i$, $K_i^*$ is its degree ranking in the network; D is a constant; and $a$ is the slope of the degree distribution curve.

(3) Agglomeration: clustering coefficient.

The clustering coefficient measures the degree of closeness of the node groups in the network. The average clustering coefficient reflects the average level of agglomeration of all nodes. The local clustering coefficient, $C_i$, is the ratio of the actual number of edges that exist between the neighbors of node i to the maximum possible number of edges that could exist between its neighbors [54]. The formula is as follows:

$$C_i = \frac{2E_i}{K_i(K_i-1)}$$
(8)

where $E_i$ is the number of edges actually generated between node $i$ and its neighboring nodes and $K_i$ is the number of neighbors of node $i$.

(4) Transmissibility: average path length, network efficiency.

The average path length, $L_{ij}$, is used to assess the network transmission efficiency, which indicates the number of nodes that the elements need to pass through in the process of flowing from node $i$ to node $j$[55]. The larger its value, the lower the propagation and diffusion efficiency and the more efficient the network operation. The formula is as follows:

$$L_{ij} = \frac{2\sum_{i>j} d_{ij}}{N(N-1)}$$
(9)

where $d_{ij}$ is the distance from node $i$ to node $j$, and $N$ is the number of network nodes.

The network efficiency, E, can measure the transmission efficiency, where $0 \leq E \leq 1$ [55]. The larger the value of E, the higher the network efficiency and the more mobile and transmissible the resources and people in the network, which is calculated by the formula:

$$E = \frac{2}{N(N-1)} \sum_{i \neq j \in G} \frac{1}{D_{ij}}$$
(10)

where $D_{ij}$ is the shortest path between node $i$ and node $j$ in the network, $G$ is the set of network nodes after removing a particular node, and $N$ is the number of network nodes.

(5) Connectivity: average degree of the network, relative size of the largest connected subgraphs.

The term 'connectivity' is used to describe the degree of access that nodes have to one another to interact. When some nodes are disturbed, a network with better connectivity can maintain its structure's stability through alternative paths. The average network degree, represented by the symbol $\overline{D}$, is used to describe the average number of nodes connected to each node [56]. In a disruption scenario, the smaller the change of $\overline{D}$, the better the connectivity. The formula is as follows:

$$\overline{D} = \frac{1}{N} \sum_{i=1}^{N} D_i$$
(11)

where $N$ is the number of network nodes, and $D_i$ is the degree centrality of the node.

The size of the maximum connected subgraph indicates the number of nodes contained in the maximum subgraph during network fragmentation [57]. Its size is denoted by G. The larger G is, the more stable and robust the network is, which is calculated as:

$$G = \frac{P^*}{P}$$
(12)

where $P^*$ is the maximum connected subgraph size, and $P$ is the initial network size.

# 3. Analysis of the results

## 3.1 Analysis of the spatiotemporal dynamics of the population flow networks

### 3.1.1 General characteristics of population flow network.

Based on the average daily population flow intensity between cities throughout the year, the migration among cities was divided into five levels, from which the connecting edges were mapped. Based on the value of composite centrality, the importance of cities was divided into three levels and nodes were plotted. A map of the spatial pattern of population flow (Fig 3) was drawn to reveal the topological hierarchies and interaction gradients in the urban network.

The overall city network is characterized by a zigzag pattern, with the main corridors being Nantong-Shanghai-Nanjing-Suzhou-Jiaxing-Hangzhou-Jinhua. The spatial distribution of the strength of connections is characterized by "dense in the middle and sparse in the north and south", with the dense connections mainly surrounding the four major cities of Shanghai, Suzhou, Hangzhou and Nanjing in a diffuse structure. Clusters tend to form between large cities across provinces (e.g., Shanghai and Suzhou, Shanghai and Hangzhou), between large cities and small and medium-sized cities in neighboring provinces (e.g., Shanghai and Jiaxing, Nantong), and between neighboring cities within the same province (e.g., Hangzhou, Huzhou, Shaoxing, Jiaxing, Jinhua in Zhejiang Province, and Nanjing, Zhenjiang, Changzhou, Yangzhou, Huai'an in Jiangsu Province). The linkage dynamics between small and medium-sized cities across provinces are obviously insufficient, indicating that administrative boundaries still have some obstacles to the development of integration, and this obstacle mainly exists between small and medium-sized cities with weak comprehensive strength, resulting in a low level of synergy in the deployment of resources, competitive relations and other aspects. The linkage between the northern and southern parts of the network is dominated by the low-level linkage, which shows that the core of regional resource factor flow is still concentrated in the economic hinterland of metropolitan areas, including Hangzhou, Nanjing, Shanghai and so on.

The entire network contains 300 interconnected pairs of cities. There are exchanges between every city, but the connection intensities vary significantly among different cities. Among all the pairs, Shanghai-Suzhou ranks first with an absolute lead (Table 1), accounting for about 8% of the total linkage intensity, and the linkage intensity is about twice that of the second-ranked city pair. It shows a clear trend of "Shanghai-Suzhou Integration", where the frequency and efficiency of cross-city factor flows have been accelerated in recent years due to the development of Shanghai-Suzhou linkages in transportation, industry and people's livelihood. Notably, the top 25 city pairs account for only 8% of the sample size, but their total linkage intensity accounts for more than 50% of the total linkage intensity of the entire network, constituting the main linkage routes for regional Population flow.

### 3.1.2 Dynamics of population flow during different periods.

To explore the relationship between population flow and temporal factors other than geospatial factors, the Spring Festival, which has a large scale of population flow and is highly cyclical, was selected as a representative period. The migration data from January 7 to February 15, 2023, were collected around the time specified by the Ministry of Transport and Communications of China. November, which has no special holidays, was selected to represent the daily period. The top 10 city pairs ranked by the average daily population migration index during different periods were counted (Table 2), and the spatiotemporal characteristics of population flow during the Spring Festival (Fig 4a) and the daily period were analyzed (Fig 4b). The top 10 city pairs in terms of migration intensity in different periods are the same objects: cities with strong comprehensive strength or outstanding industrial characteristics, such as e-commerce powerhouses Hangzhou and Jinhua. It indicates that these cities have a long history of high-density economic exchanges and industrial linkages and that the intensity of intercity migration for daily commuting is higher than that for returning to work on specific holidays, which slows down due to the work stoppage during the Spring Festival. The number of city pairs varied across the levels of connectivity. It is most evident in the fact that the number of city pairs in the second tier increased by one pair in the daily period compared to the Spring Festival, while there were 24 pairs in the third tier in the daily period and 33 pairs in the Spring Festival; 48 pairs in the fourth tier in the daily period and 64 pairs in the Spring Festival; and the number of pairs in the fifth tier did not change much. It is worth

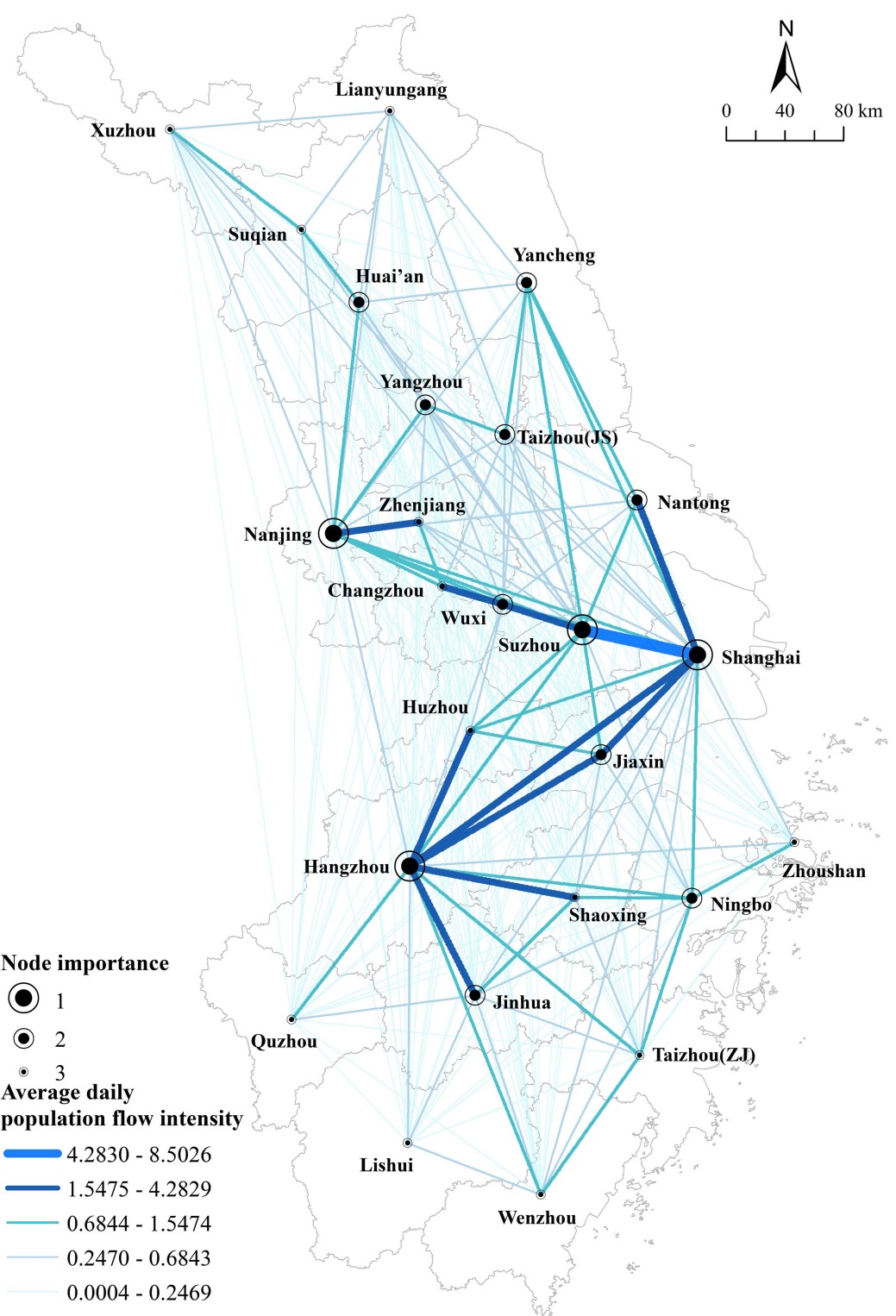

**Fig 3. Spatial patterns of population flow over the year.** Reprinted from [(http://www.resdc.cn/DOI),2023.DOI:10.12078/2023010101] under a CC BY license, with permission from [Xu Xinliang], original copyright [January 2023].

**Table 1. Top 25 city pairs in terms of intensity of population flow in one year.**

| Rank | City pair | Average daily migration index | Share of total linkage intensity |
|------|-----------|-------------------------------|----------------------------------|
| 1 | Suzhou-Shanghai | 8.50 | 8.17% |
| 2 | Jiaxing-Hangzhou | 4.28 | 4.11% |
| 3 | Shaoxing-Hangzhou | 3.87 | 3.72% |
| 4 | Wuxi-Suzhou | 3.62 | 3.48% |
| 5 | Wuxi-Changzhou | 2.80 | 2.69% |
| 6 | Huzhou-Hangzhou | 2.62 | 2.52% |
| 7 | Zhenjiang-Nanjing | 2.60 | 2.50% |
| 8 | Shanghai-Hangzhou | 2.31 | 2.22% |
| 9 | Shanghai-Jiaxing | 2.21 | 2.12% |
| 10 | Jinhua-Hangzhou | 2.00 | 1.93% |
| 11 | Shanghai-Nantong | 1.97 | 1.89% |
| 12 | Ningbo-Hangzhou | 1.55 | 1.49% |
| 13 | Wuxi-Shanghai | 1.38 | 1.32% |
| 14 | Suzhou-Nantong | 1.36 | 1.30% |
| 15 | Zhenjiang-Hangzhou | 1.31 | 1.26% |
| 16 | Shaoxing-Ningbo | 1.28 | 1.23% |
| 17 | Suzhou-Jiaxing | 1.16 | 1.12% |
| 18 | Yangzhou-Nanjing | 1.10 | 1.05% |
| 19 | Shanghai-Nanjing | 1.05 | 1.01% |
| 20 | Suzhou-Nanjing | 1.03 | 0.99% |
| 21 | Taizhou(ZJ)-Ningbo | 1.01 | 0.97% |
| 22 | Xuzhou-Suqian | 1.01 | 0.97% |
| 23 | Shanghai-Ningbo | 1.00 | 0.96% |
| 24 | Wenzhou-Taizhou(ZJ) | 0.97 | 0.93% |
| 25 | Yangzhou-Taizhou(JS) | 0.93 | 0.89% |

**Table 2. Top 10 city pairs in terms of intensity of population flow during the Spring Festival and daily period.**

| Rank | | City pair | Average daily migration index | | City pair | Average daily migration index |
|------|----------------|-----------|-------------------------------|--------------|-----------|-------------------------------|
| 1 | Spring Festival | Suzhou-Shanghai | 6.18 | Daily period | Suzhou-Shanghai | 8.22 |
| 2 | | Shaoxing-Hangzhou | 3.09 | | Jiaxing-Hangzhou | 4.12 |
| 3 | | Jiaxing-Hangzhou | 2.86 | | Shaoxing-Hangzhou | 3.68 |
| 4 | | Wuxi-Suzhou | 2.58 | | Wuxi-Suzhou | 3.51 |
| 5 | | Zhenjiang-Nanjing | 2.23 | | Changzhou-Wuxi | 2.74 |
| 6 | | Changzhou-Wuxi | 2.16 | | Hangzhou-Huzhou | 2.52 |
| 7 | | Huzhou-Hangzhou | 1.94 | | Zhenjiang-Nanjing | 2.47 |
| 8 | | Jinhua-Hangzhou | 1.91 | | Jiaxing-Shanghai | 2.08 |
| 9 | | Nantong-Shanghai | 1.90 | | Hangzhou-Shanghai | 1.86 |
| 10 | | Hangzhou-Shanghai | 1.69 | | Jinhua-Hangzhou | 1.73 |

noting that the changes in city pairs in each tier are not simply an increase in the number of pairs; their objectives are also different. For example, in the third tier of city pairs, the closer daily links between Hangzhou and Suzhou, Shanghai and Huzhou, Jiaxing and Huzhou, and Jinhua and Shaoxing all saw a significant decrease in their links during the Spring Festival, confirming what was said above that some of the more developed cities are closer to each other in terms of daily

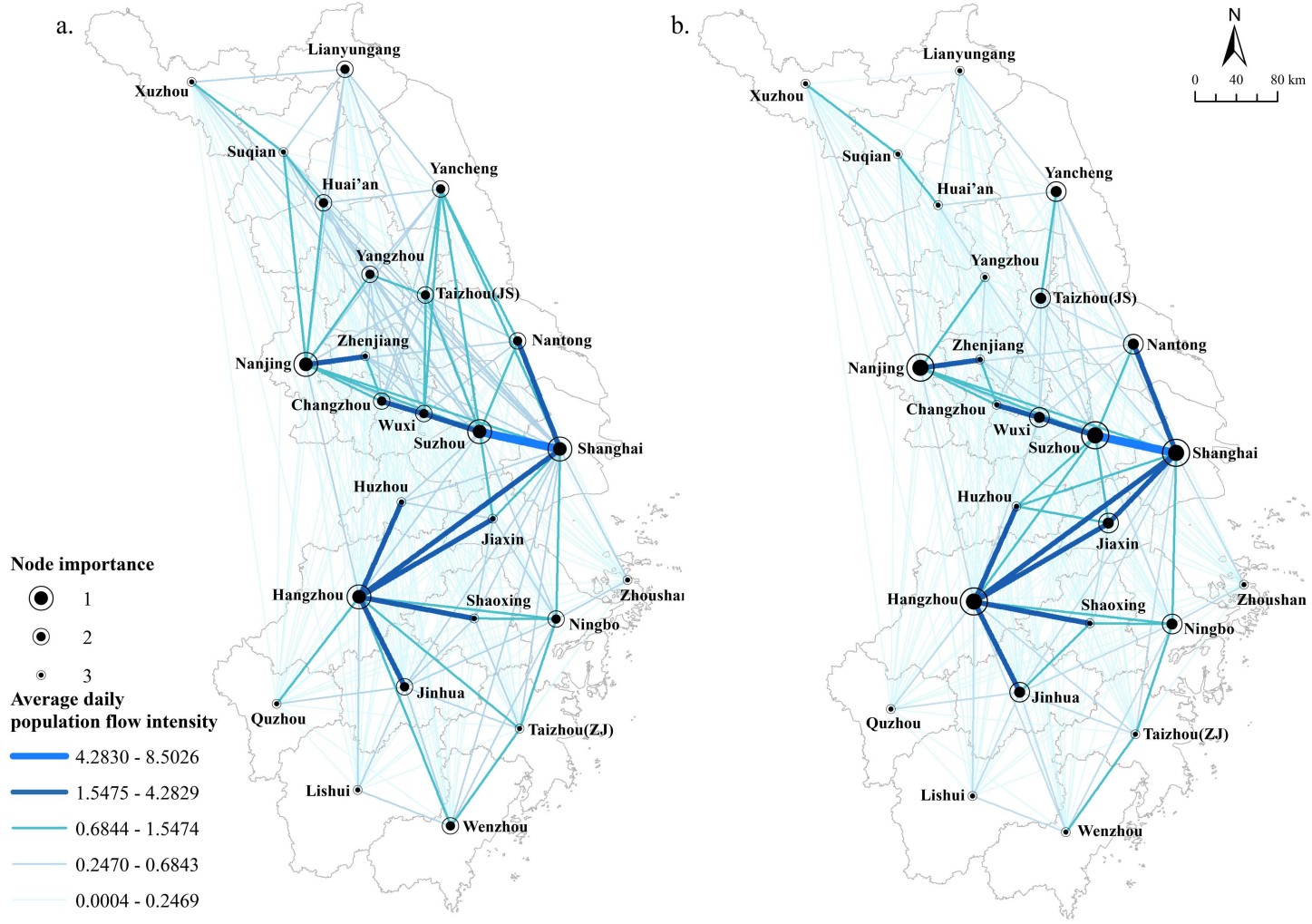

**Fig 4. Spatial patterns of population flow during the Spring Festival (a) and daily period (b).** Reprinted from [(http://www.resdc.cn/DOI),2023. DOI:10.12078/2023010101] under a CC BY license, with permission from [Xu Xinliang], original copyright [January 2023].

commuting and economic interactions. A comprehensive comparison of the different periods shows that the intensity of city network connections during the Spring Festival is significantly higher than that of the daily period, and the increased connections of the third and the fourth tiers during the Spring Festival are mainly distributed between developed cities and other cities within the same province, suggesting that the phenomenon of cross-city commuting or migration for work is more pronounced among the population of the study area.

## 3.2 Assessment of urban network structure resilience

### 3.2.1 Analysis of the network structure resilience under normal scenario.
Network centrality. To assess the importance of nodes within the network, degree centrality, betweenness centrality and closeness centrality are applied, and processed to get the composite centrality (Fig 5). These four indicators have apparent spatial heterogeneity and can comprehensively reflect the network centrality. Analysis shows that Shanghai, Suzhou, Hangzhou and Nanjing are among the top 4 cities regarding various key indicators. Also, they are the core cities with leading status in the network,

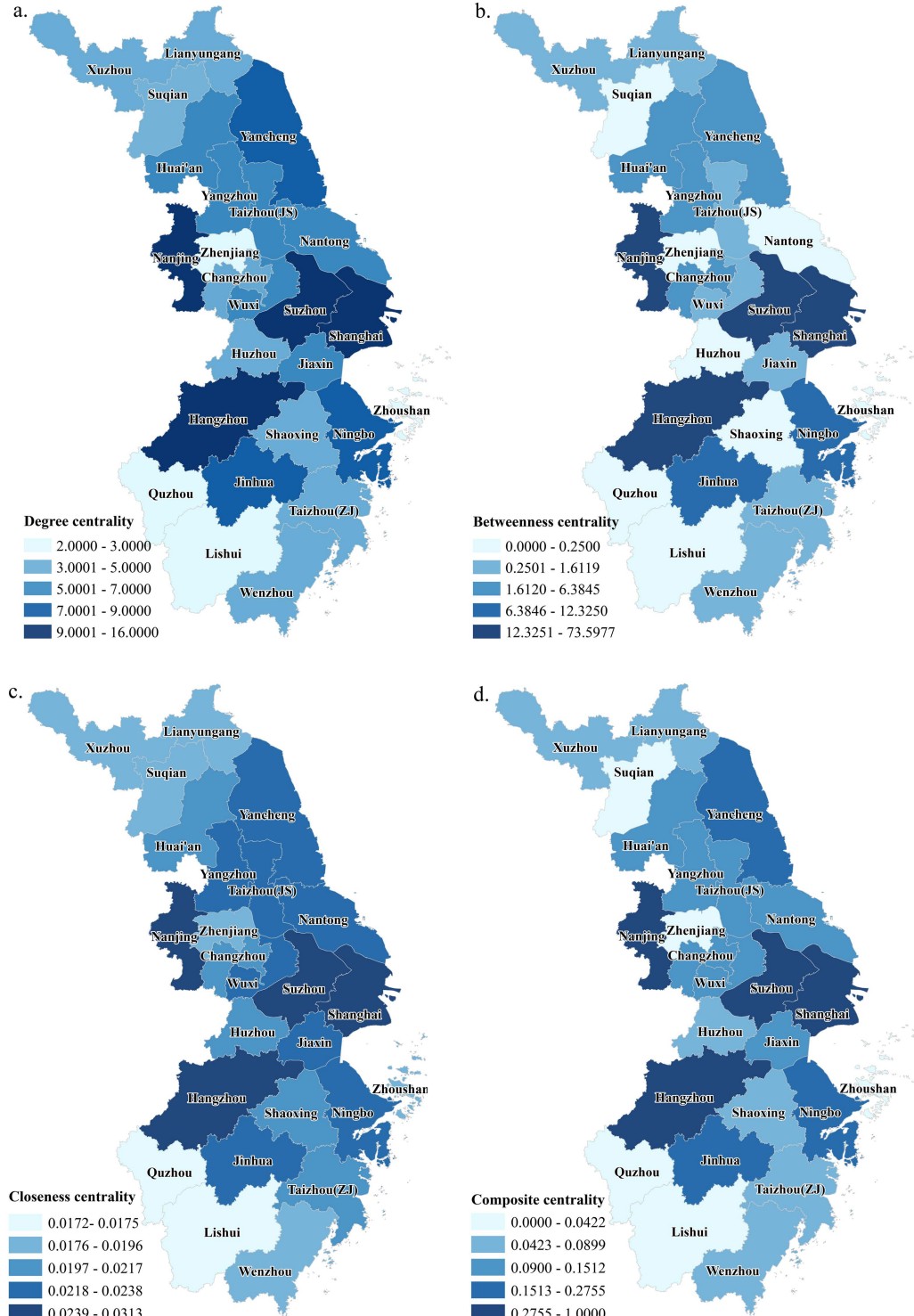

**Fig 5. Spatial distribution of network degree centrality(a), betweenness centrality(b), closeness centrality(c), and integrated centrality(d).**
Reprinted from [(http://www.resdc.cn/DOI),2023.DOI:10.12078/2023010101] under a CC BY license, with permission from [Xu Xinliang], original copyright [January 2023].

with strong radiation, driving and connecting effects on other cities. Jinhua, as the "No.1 express delivery city in China", Ningbo, as a sea and land transportation hub, and Yancheng, as a significant agricultural city and an emerging science and technology city, are nodes of high interest in the whole network; they serve as an important link in the flow of factors, and the average distance between them and other nodes is small, making them attractive to other cities and having strong radiating ability. Quzhou and Lishui are at the bottom of the list in terms of each type of centrality, and Zhenjiang is located in the middle of the region and close to the core city of Nanjing, but its centrality in each category is low; this indicates that these cities have insufficient comprehensive strength and competitiveness, and they play a weaker role in regional interactions. Overall, the centrality of the central part in this region is significantly higher than that in the northern and southern parts. Moreover, the cities in the periphery, such as Xuzhou, Lianyungang, Suqian, Quzhou, Lishui, and Wenzhou, are far from the agglomeration areas of the core cities, lacking efficient links with developed cities and cooperation mechanisms and having single-city functions and low factor flow efficiency.

Network hierarchy. Linear fitting with degree centrality yields a degree distribution index of −0.5184 (Fig 6), which still has a larger gap from 1. Viewed from the angle of resilience, the hierarchical structure of the network lacks sufficient prominence, giving rise to relatively feeble overall regional competitiveness. However, the absolute value of the degree distribution index is more significant than 0.5, indicating that the distribution hierarchy of the urban linkage network in the study area has a certain degree of rationality. The gap between the degree centrality of core cities such as Shanghai, Suzhou, Hangzhou, Nanjing, and other cities is relatively small, and the network development is relatively balanced. The linkage paths of each node are more diversified, and the failure of a node has a relatively limited impact on the normal operation of the overall network, which can, to a certain extent, guarantee the efficient transmission and diffusion of various elements and help maintain the resilience of the network structure.

Network agglomeration. The average clustering coefficient within the network attains a value of 0.725, which indicates that the clustering effect of the network is more obvious, and most of the nodes in the network are well-connected and have fewer islands. In terms of the local clustering coefficient (Fig 7), the local clustering coefficients of the four core cities of Shanghai, Suzhou, Nanjing and Hangzhou are all lower than 0.5, which indicates that the interaction between the non-core cities connected to these cities is not close. There are more unidirectional links between non-core cities and core cities in the network, but there is a lack of full interaction and cooperation among small and medium-sized cities

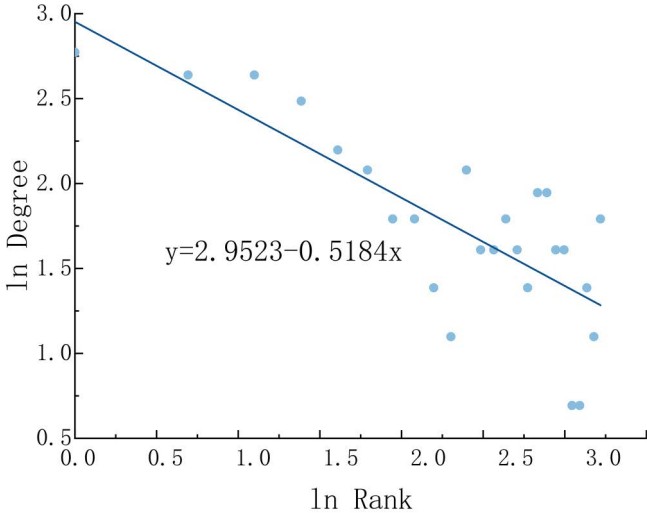

**Fig 6. Degree Distribution.**

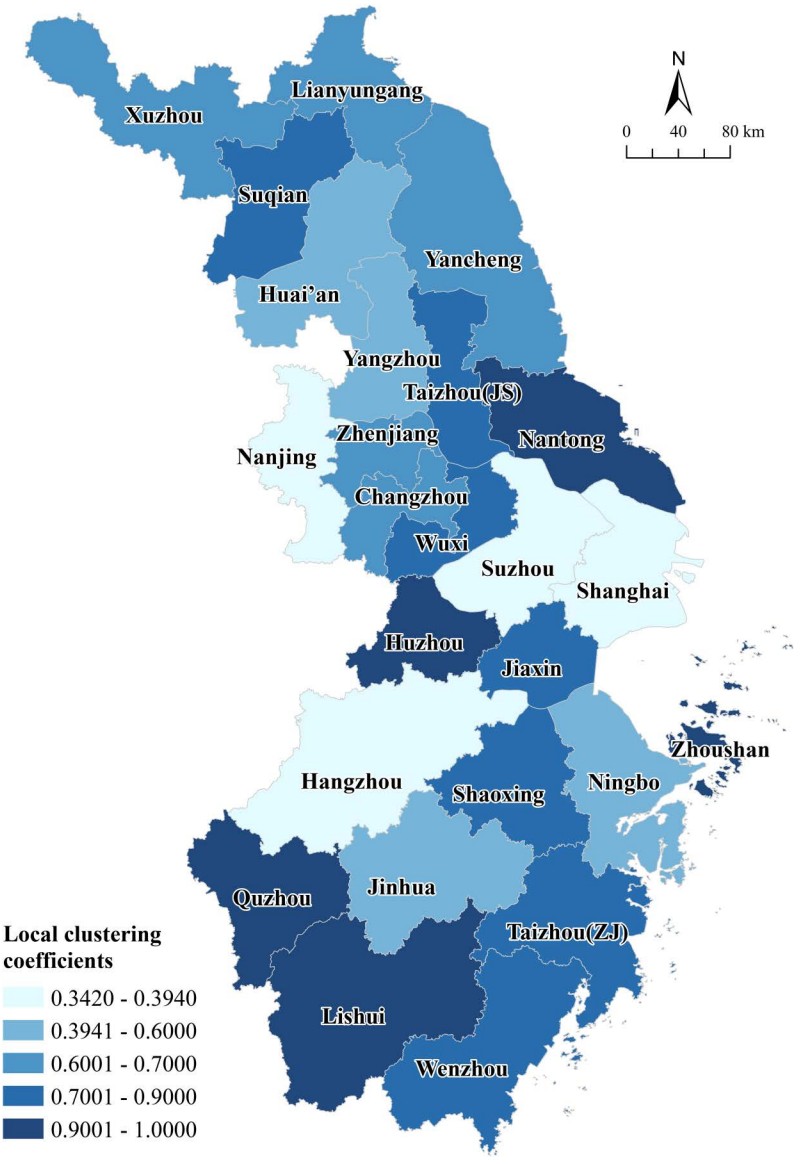

**Fig 7. Spatial distribution of local clustering coefficients.** Reprinted from [([http://www.resdc.cn/DOI),2023.DOI:10.12078/2023010101](http://www.resdc.cn/DOI)] under a CC BY license, with permission from [Xu Xinliang], original copyright [January 2023].

in the non-core status. The local clustering coefficients of Huzhou, Lishui, Nantong, Quzhou and Zhoushan all reach 1, indicating that these cities are more firmly connected with their neighboring cities, and their external service functions and competitiveness are not strong. From the perspective of network structural resilience, moderate overall agglomeration is conducive to the formation of a more sustainable cooperative relationship between nodes with loose and moderate connections. In addition, the linkages between cities are not too dense or consolidated, which can avoid the formation of information barriers and provide more opportunities for information flow within and outside the network, thus making the network more resilient.

Network transmissibility. The average path length of the network is 1.937, and the element transmission can be realized after no more than two transit nodes between nodes. The space-time cost of element exchange is low, and the network

transmission efficiency is high. Specifically (Fig 8), 83 paths in the network do not need intermediaries to directly generate connections, accounting for 27.67%; 153 paths can be connected by only one transit, accounting for 51.00%. The analysis shows that nearly 80% of the cities can be directly connected within only one transit at most, the network accessibility and diffusivity are relatively high, and activities such as personnel flow, technology exchange and information sharing between regions usually run efficiently. The efficiency of regional resource integration and distribution is high. Overall, the network transmissibility is good, and the efficient and low-cost dissemination is conducive to ensuring the network's adaptability and resilience during a crisis. In addition, 21.33% of the paths must pass through two transits before the city can establish a connection. The accessibility and transmission efficiency of such city nodes are not high, which is a weak part of urban network resilience, and the interference of their intermediate nodes will quickly lead to local paralysis of the network, thus limiting the improvement of the resilience level of the overall network structure.

**3.2.2. Characteristics of changes in network structural resilience under disruption scenario simulation.** The urban network of population flow consists of node cities and connectivity paths, and changes in node function and path structure due to disruption will definitely affect the network resilience level to different degrees (Fig 9). Overall, both random disruption and targeted disruption exert a marked influence over network resilience. The high convergence of the changing patterns of transportability and connectivity indicates that during a crisis, the number and the length of paths, the operational efficiency, and the tightness of node connectivity of intercity population flow will be affected, and the difficulty of intercity factor mobility will be increased. As a result, the ability of the network structure to respond to emergencies, risk

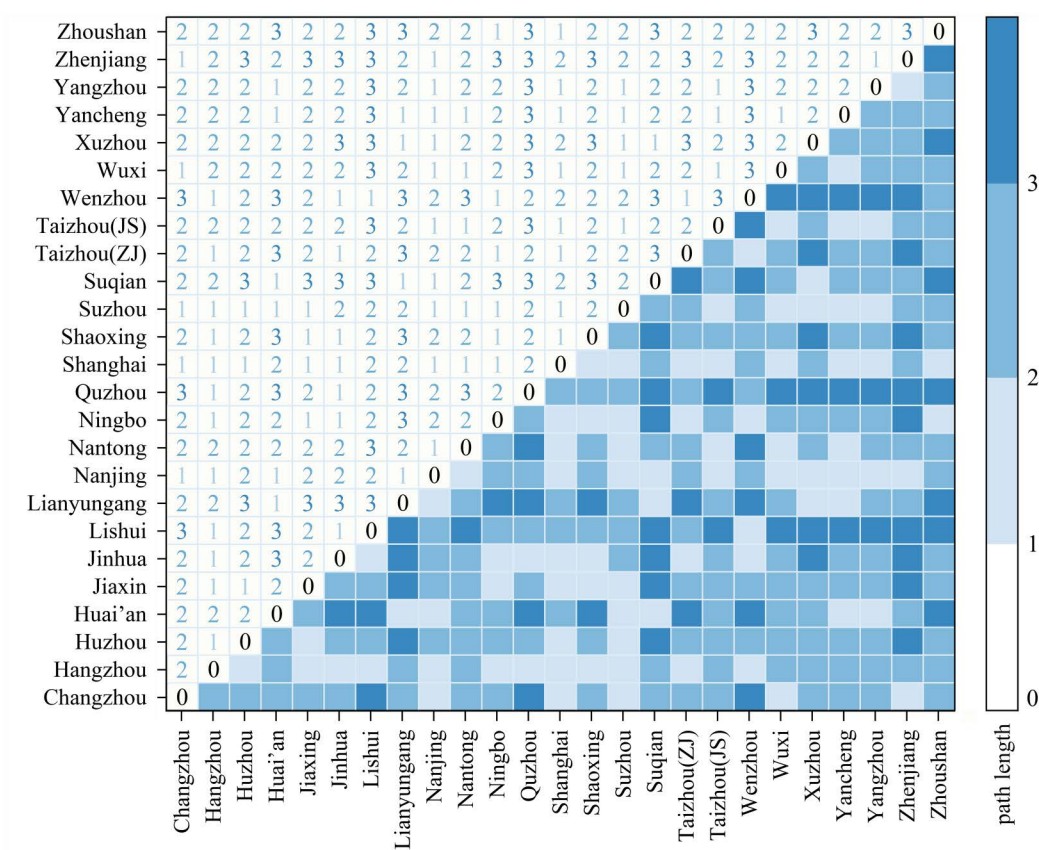

**Fig 8. The shortest path length between network nodes.**

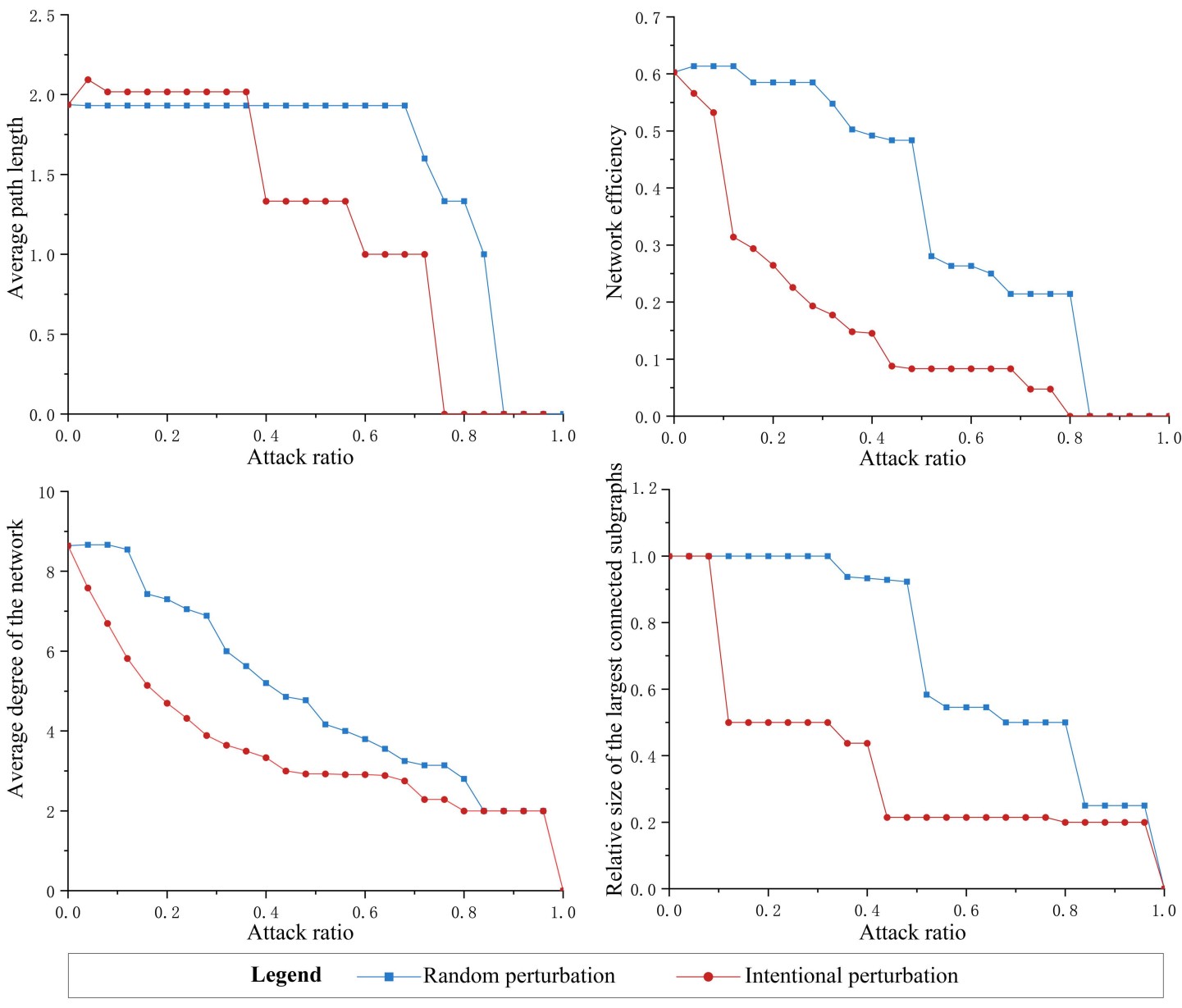

**Fig 9. Changes of network structure resilience under disruption scenario simulation.**

resilience, and post-disaster recovery will be weakened, and this series of chain reactions will exacerbate the vulnerability of the network and degrade the robustness grade of the network structure. Furthermore, the network experiences phase transitions under both kinds of disruptions, but the magnitudes of the changes differ. Targeted disruption can substantially impair the network's resilience level more severely than random disruption, leading the network to succumb to failure at an earlier stage. Critical points occur under both disruptions, causing the metrics to change drastically. The disruptions initially have a negligible effect on the resilience of the network, but when they reach a critical value, the resilience level drops rapidly, and the network collapses. The network under targeted disruption with a node attack ratio of 70% or so collapses rapidly, and the network under random disruption with an attack ratio of 80% or so collapses violently. It shows

a core network of a few key nodes in the population flow network in the study area, which can maintain a complete functional structure to support the operation until it is severely perturbed.

Specifically, the network resilience changes significantly under targeted disruption. The average path length first increases briefly and then maintains a stable value slightly higher than the standard value until the proportion of attacks reaches 36%, and then starts to decrease abruptly; at this stage, the network efficiency and the average degree are linearly reduced, and the relative size of the maximum connectivity subgraph fluctuates, and the network resilience generally undergoes a steady change, but it can still maintain some degree of regular operation. When the attack ratio reaches 56%, the average path length decreases again, and a large number of nodes are removed, causing severe damage to the network connectivity, while the rate of change of the average path length, average degree, and relative size of the largest connected subgraph slows down or even flattens, indicating that the overall network fragmentation is more pronounced. When the attack ratio increases to 72%, the average path length drops to 0, the network is basically disintegrated, and the nodes tend to be isolated.

In terms of changes in network resilience under targeted disruption, significant changes in the relevant metrics occur when the percentage of attacks reaches 36% and 56%, so these two thresholds are used to identify the core nodes that are critical to the overall network. The first 36% of nodes and their connections constitute the core network part with critical impact, and the first 56% of nodes and their connections constitute the more holistic network part that can maintain the network structure and function completely (Table 3). The core network nodes are usually more substantial in terms of overall strength and have a more significant radial influence on neighboring cities in terms of development orientation and resource allocation, and failure of the nodes will highly constrain the efficient operation of the network. The stable development of the node cities in a more complete network can guarantee the regular operation of the region. The remaining nodes are regarded as the peripheral nodes within the network, exerting a relatively minor influence on the index value associated with the network's regular operation. These nodes still have a large gap with developed cities in terms of transportation infrastructure, public service facilities, and job provision. Moreover, they have low interrelated development with neighboring cities and a low regional influence.

## 4. Discussion

### 4.1 Research significance and characteristics of population flow network resilience

Current resilience research lacks a unified theoretical framework and systematic evaluation metrics. This study develops a research framework of "structural analysis-scenario simulation-resilience assessment" to systematically investigate the characteristics of population flow network resilience. Through normal and disruption scenarios, urban network resilience response mechanisms were revealed. This framework establishes a significant methodological paradigm for analyzing network structural features and measuring resilience. Unlike traditional static topological analyses that fail to capture the cascade effects of node failures [58], the dynamic disruption simulation quantitatively evaluates network transmissibility and connectivity, elucidating the evolving impacts of node failures on overall structural resilience. These findings align with existing literature, yet we further identify the dual "hub-and-bottleneck" roles of core nodes in risk propagation, whose topological attributes amplify cascade effects. This constitutes one of the characteristic patterns in the network evolution of urban clusters and metropolitan areas [59,60].

Table 3. Core network, relatively complete network and edge node identification.

| Core network | Relatively complete network | Edge node |
|---|---|---|
| Shanghai, Nanjing, Suzhou, Hangzhou, Ningbo, Jinhua, Yancheng, Wuxi, Taizhou(JS) | Shanghai, Nanjing, Suzhou, Hangzhou, Ningbo, Jinhua, Yancheng, Wuxi, Taizhou(JS), Yangzhou, Nantong, Huai'an, Jiaxing, Changzhou | Xuzhou, Lianyungang, Suqian, Zhenjiang, Huzhou, Zhoushan, Shaoxing, Quzhou, Taizhou(ZJ), Lishui, Wenzhou |

Methodologically, many studies commonly use single-dimensional centrality indicators, such as only using degree centrality to assess the influence of cities [61], or using degree centrality and betweenness centrality to identify transportation hubs [62]. In contrast, this study comprehensively applies degree centrality, betweenness centrality, and closeness centrality to construct the composite centrality, providing a multi-dimensional perspective to fully characterize the role of cities. The results of targeted disruption show that the failure of cities with higher composite centrality will trigger a "gradient decline in resilience", while the failure of non-core nodes will only lead to local paralysis. Specifically, removing the top 20% of core cities can reduce network resilience by more than half, while random disruption requires the removal of approximately 50% of the nodes to achieve a similar level of damage. This indicates that networks relying on core nodes are efficient but have high vulnerability. This study provides a quantitative tool for urban risk prevention and control and enriches the analysis of disaster chain networks [63].

Structurally, the Jiangsu-Zhejiang-Shanghai region exhibits a "central densification with northern and southern sparsity" pattern, consistent with regional studies [64]. Further analysis reveals that the connection strength between developed cities during the daily period is higher than that during the Spring Festival. For example, the average daily flow intensity between Shanghai and Suzhou on working days is 33% higher than that during the Spring Festival. This finding challenges the classical festival-dominant migration paradigm [65] and likely stems from high-frequency flows engendered by routine economic interactions such as industrial collaboration and metropolitan commuting. This indicates an advanced stage of economic integration in the study area, wherein population mobility transitions from cyclical holiday-driven movements to market-sustained regular exchanges. Furthermore, the top 25 city-pairs in terms of connection strength account for only 8% of the sample size but contribute more than 50% of total network connectivity. This conforms to the power-law distribution in complex network theory [66]. This polarization feature enhances the efficiency of resource allocation while also implying regional network vulnerability. The study also found that administrative boundaries have a more significant inhibitory effect on the connections of small and medium-sized cities than on those of large cities, expanding Wang's discussion on the impact of administrative divisions [67]. Based on the above research results, this study proposes a "resilience-oriented" regional policy design path, providing empirical evidence for breaking through administrative boundary barriers.

## 4.2 Policy implications

The concept of resilience serves as the foundational principle for the well-structured network, which is designed to provide effective support for the maintenance of security, stability, and quality development in the region. The deductions drawn from the aforementioned research bear considerable referential value for devising city layout and spatial evolution strategies within the region. Policy recommendations are as follows:

**4.2.1 Strengthen the leading effect of core cities.** From the perspective of the overall regional structure, the urban network hierarchy is relatively low, implying that the radiation scopes of the core cities are relatively circumscribed. Under the disruption scenario, the core network is composed of nine cities, which should be taken as powerful leaders in the future. On the one hand, strengthen the "core-core" synergistic linkage and enhance the radiation power of core cities as distribution points for regional factor flows. A joint governance mechanism for resilience corridors such as Shanghai-Hangzhou-Nanjing-Suzhou should be established, with a focus on enhancing the hub functions of the nine core cities in cross-provincial collaboration and joint prevention of crises. On the other hand, the effectiveness of "core-periphery" connectivity should be improved. For weak radiation belts such as Nanjing-Zhenjiang and Hangzhou-Jinhua, targeted improvement policies of building joint industrial enclaves and joint training of talents should be implemented to create competitive development clusters. Through a multi-level, strongly linked city network system, the overall development quality and risk-resistant capacity of the region will be effectively enhanced.

**4.2.2 Enhance the development of sub-resilience nodes.** Based on an analysis of the spatio-temporal characteristics and resilience of the network, there are connectivity deficits in northern Jiangsu and southern Zhejiang provinces, indicating

low mobility and weak network influence. The region's comprehensive competitiveness is significantly lower than that of the developed central area. To optimize the urban network structure, regional sub-centers should be cultivated based on resource endowment. Xuzhou should leverage its industrial foundation to attract spillover industries from core cities and promote the synergistic development of neighboring cities, such as Lianyungang and Suqian. At the same time, by leveraging the ecological and cultural resources of Quzhou and Lishui, these cities can attract a population shift from the Hangzhou Metropolitan Area and achieve a dynamic balance in the regional mobility network. Upgrading the secondary cities' comprehensive carrying capacity in terms of industrial support and service sharing promotes the transformation of population mobility patterns from core agglomeration to a multi-center network and builds a more resilient regional spatial pattern.

**4.2.3 Optimize factor transmission paths.** Seen from the perspective of the element flow trend within the network, numerous non-core cities in proximity to provincial administrative boundaries, despite being geographically adjacent, display relatively low levels of element connectivity. Therefore, it is imperative to dismantleadministrative border barriers and build a platform for monitoring factor flows, to identify "invisible barriers" to mobility promptly. In addition, an intercity resource-sharing system should be piloted, focusing on breaking through institutional barriers in key areas such as healthcare, education and emergency resources. Under the impetus and radiation of core cities, horizontal compensation networks between subnodes should be improved, thus promoting the efficient flow of elements through the pathways at all echelons of the network space. These measures will effectively enhance the network participation of edge nodes and optimize factor allocation in the region.

## 4.3 Limitations and future directions

This study advances urban resilience research through a multidimensional analytical framework based on population migration flows, but there are still some limitations. The analysis of the mobile population is limited to the characteristics of quantity and travel time and has not included attribute dimensions such as age, occupation, and education level, making it difficult to analyze the differentiated impact of group heterogeneity on network resilience. Regarding spatial scale, the research focuses on the relationship within the region and has not yet built a multi-level resilience assessment system for the "urban agglomeration-national-global" hierarchy . The transmission mechanism of external disturbance factors needs to be explored. Theoretically, the existing models reveal urban relations through static and dynamic network topologies. In the future, we can integrate game theory frameworks to model the long-term resilience effect of urban competitive and cooperative strategies, which will focus on analyzing network resilience under extreme flow situations, such as special periods and events. These research directions will greatly expand the depth and breadth of future research.

## 5. Conclusions

Taking the urban network of population flow in the Jiangsu-Zhejiang-Shanghai region of China as an object, this study analyzes the spatiotemporal dynamics of urban linkages throughout the year, typical festivals and daily periods. The characteristics and response mechanisms of urban network resilience are revealed through structural analysis, scenario simulation and resilience assessment. The results of the study are of great significance in guiding the optimization of regional spatial patterns and the formulation of development policies. The main conclusions are as follows:

Regarding spatial distribution, the network connections are dense in the center and sparse in the north and south. Overall, there is a zigzag-shaped main corridor centered on Shanghai, Suzhou, Hangzhou, and Nanjing, with a small number of high-level networks accounting for more than half of the total contact intensity. Clusters are more likely to form between developed cities across provinces or between neighboring cities in the same province, and the obstacles to regional integration posed by administrative boundaries are more pronounced in the links between small and medium-sized cities. In the time dimension, the number of city pairs at each level of linkage intensity varies during different periods, and the phenomenon of population commuting across cities or going out to work is evident. The overall linkage intensity of the city network during the Spring Festival is significantly higher than that of the daily period. Among

them, the city pairs with the highest migration indexes have a higher daily migration intensity than those during the Spring Festival, which suggests that the daily commuting and economic exchanges between developed cities are more frequent.

Under the normal scenario, the centrality of the northern and southern parts of the study area is significantly lower than that of the central part, and the centrality of the four core cities of Shanghai, Suzhou, Nanjing, and Hangzhou is the most prominent. Network hierarchy is not significant, and the comprehensive strength of the region needs to be strengthened. Network agglomeration is good, and the loose linkage is favorable to the free flow of factors. Network transmission efficiency is high, with more than 78% of the cities connected by at most one link, and the time and space cost of factor exchange is low. The linkage between core cities and non-core cities is asymmetric, and the interaction between small and medium-sized cities is insufficient, resulting in the edge cities being in a passive succession position in factor transmission.

Under the simulation of the disruption scenario, the changing dynamics of network transportability and connectivity have a high degree of convergence, and the network undergoes phase changes. In addition, targeted disruption significantly reduces network resilience more than random disruption. A small number of nodes with critical influence form the core network, and half of the nodes form the more complete network. Targeted enhancement of the toughness of a few core nodes can substantially improve the security and efficiency of the overall network.

## Supporting information

**S1 Data. Data file and availability statement.**
(ZIP)

## Author contributions

**Conceptualization:** Jiulin Li, Wenhui Lin.

**Data curation:** Wenhui Lin.

**Formal analysis:** Wenhui Lin.

**Funding acquisition:** Jiulin Li.

**Investigation:** Wenhui Lin.

**Methodology:** Jiulin Li, Wenhui Lin.

**Project administration:** Jinlong Chu.

**Resources:** Jiulin Li.

**Software:** Wenhui Lin.

**Supervision:** Jinlong Chu.

**Validation:** Jinlong Chu.

**Visualization:** Wenhui Lin.

**Writing – original draft:** Wenhui Lin.

**Writing – review & editing:** Jiulin Li, Jinlong Chu.

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
