## [Decision Letter · Decision Letter 0]

May 12 2025

Dear Dr. Li,

Thank you for submitting your manuscript to PLOS ONE. After careful consideration, we feel that it has merit but does not fully meet PLOS ONE’s publication criteria as it currently stands. Therefore, we invite you to submit a revised version of the manuscript that addresses the points raised during the review process.

We look forward to receiving your revised manuscript.

Kind regards,

Qiwei Ma

Academic Editor

PLOS ONE

Journal Requirements:

“This research was funded by the Anhui Office of Philosophy and Social Science, grant number AHSKD2023D028.”

“This research was funded by the Anhui Office of Philosophy and Social Science, grant number AHSKD2023D028. The grant Recipient is Jiulin Li (JL), and his orcid number is 0009-0001-1967-8124. He was  responsible for conceptualization, funding acquisition, methodology, resources, visualization and writing– review & editing.”

Additional Editor Comments (if provided):

Based on the comments of reviewers, I have reached the decision: Major Revision.

Reviewers' comments:

Reviewer's Responses to Questions

**Comments to the Author**

1. Is the manuscript technically sound, and do the data support the conclusions?

Reviewer #1: Yes

Reviewer #2: Yes

Reviewer #3: Yes

2. Has the statistical analysis been performed appropriately and rigorously?

Reviewer #1: Yes

Reviewer #2: Yes

Reviewer #3: Yes

3. Have the authors made all data underlying the findings in their manuscript fully available?

Reviewer #1: Yes

Reviewer #2: Yes

Reviewer #3: Yes

4. Is the manuscript presented in an intelligible fashion and written in standard English?

Reviewer #1: Yes

Reviewer #2: Yes

Reviewer #3: Yes

Reviewer #1: This manuscript explores the spatiotemporal dynamics and resilience assessment of urban networks in the Yangtze River Delta region, based on complex network theory. The topic is of significant relevance, and the methodology is generally sound. However, there are a few areas that need further clarification to enhance the scientific rigor and credibility of the paper. The specific review comments are as follows:

1. Clarification on the Origin and Calculation of Network Metrics: In lines 229-268, several network metrics are introduced, including clustering coefficient, average path length, network efficiency, average degree of the network, and relative size of the largest connected subgraphs. However, the paper does not provide citations for the origin of these metrics. If these metrics are proposed by the authors, it is important to elaborate on their calculation methods and provide theoretical justification to demonstrate their credibility and relevance for assessing network efficiency.

2. Inconsistency between Table 2's Title and Content: The title of Table 2 is “Top 25 city pairs in terms of intensity of population flow during the Spring Festival and daily period,” but the table only lists 10 pairs of cities for each of the two periods. The title should be revised to accurately reflect the content of the table, or the authors should explain why only 10 pairs are included.

3. Unclear Description of “Intentional Perturbation”: In section 3.2.1, the term “intentional perturbation” is mentioned, but its precise meaning and application are unclear. The paper does not specify which cities were targeted by the perturbation or provide any details on how the perturbation was implemented. It would be helpful to clarify the definition of “intentional perturbation” and provide further explanation of its implementation and impact, so that readers can better understand the research design.

Overall, the manuscript presents a valuable analysis framework, but additional details and clearer references are needed to enhance the scientific rigor of the paper.

Reviewer #2: I think the major problem of this article is the lack of innovation. Many studies have used population flow data to construct urban networks and analyze their structure and resilience characteristics. This article is not innovative enough in terms of methodological design, selection of network structure indicators and resilience measurement, and it does not draw interesting or outstanding results. In addition, I don't see “The static characteristics of network resilience in natural scenarios were analyzed” mentioned by the author in the abstract. The authors just analyzed the structural characteristics of networks using some common network metrics. Specifically, the article has the following deficiencies:

1.There are some problems in the structural arrangement of the article. Usually, the authors should first further discuss the main findings of the study in the discussion section, such as comparing their own research methods, contents and conclusions with the current study, so as to better reflect the innovation of the article. Then in the final section of conclusion, the authors should summarize the methodology, conclusions, research significance, policy implications, and outlook for future research.

2. There are some formatting problems in the reference.

3. There are some language specification problems, for example, in line 171, the terms “betweeness centrality’ and “closeness centrality” seem to be more recognized than “mediating centrality” and “proximity centrality” used in this article.

4. There are inconsistencies in the use of the same term, e.g., whether “toughness” or “resilience’ is used for resilience?.

5.In the part 3.2.1, line 353: What is “compose centrality”? What is the basis for normalizing the three centrality indicators to obtain the so-called “compatible centrality”? The authors should provide some supporting evidence, such as references. Also, in the caption of Figure 5, “compose centrality” is written as “integrated centrality”.

Reviewer #3: The paper explores the spatiotemporal dynamics of population flow and network resilience within the Jiangsu-Zhejiang-Shanghai region using one-year Baidu daily migration scale index datasets from 2023. The topic is very interesting. There are some issues that need to be addressed.

1.The data used in the manuscript is the daily migration scale index rather than OD flows. How is the index calculated and what does its value mean?

2.In Section 2.4.2, do the authors consider travel time or other impacts that reflect the level of transportation facilities when measuring network transmission efficiency? If not, please explain the reason.

3.In Section 3.1.2, the authors just compared the differences in migration index between the spring festival and a typical month. Given the advantage of having a full year of data, a more detailed exploration of temporal flow variations would be beneficial. Besides, the results in Figure 4 are not very clear.

4.Regarding the scenario simulation in Section 3.2.1, how are random and deliberate perturbations defined and implemented?

5.There are some grammar errors and long sentences such as in Lines 77-82. The authors are recommended to revise the text to improve clarity and readability.

**Do you want your identity to be public for this peer review?** For information about this choice, including consent withdrawal, please see our Privacy Policy

Reviewer #1: No

Reviewer #2: No

Reviewer #3: No

---

## [Author Response · Author response to Decision Letter 1]

18 Apr 2025

Response to Reviewers

Dear Editors and Reviewers:

We sincerely appreciate your constructive feedback on our manuscript entitled “Exploring the Spatiotemporal Dynamics and Resilience Assessment of Urban Networks from the Perspective of Population Flow” (Manuscript Number: PONE-D-25-01272). Your expert comments have provided invaluable guidance for enhancing both the scholarly rigor and practical relevance of this study. We have meticulously addressed each suggestion through comprehensive revisions.In the following, the responses to all the comments are provided one by one.

Thank you for your time and thoughtful consideration of our work.

Response to Reviewer #1

This manuscript explores the spatiotemporal dynamics and resilience assessment of urban networks in the Yangtze River Delta region, based on complex network theory. The topic is of significant relevance, and the methodology is generally sound. However, there are a few areas that need further clarification to enhance the scientific rigor and credibility of the paper. The specific review comments are as follows:

1. Clarification on the Origin and Calculation of Network Metrics: In lines 229-268, several network metrics are introduced, including clustering coefficient, average path length, network efficiency, average degree of the network, and relative size of the largest connected subgraphs. However, the paper does not provide citations for the origin of these metrics. If these metrics are proposed by the authors, it is important to elaborate on their calculation methods and provide theoretical justification to demonstrate their credibility and relevance for assessing network efficiency.

The author’s answer: Thank you for your valuable suggestions. The relevant metrics were derived from classic literature and were previously cited improperly. The revised manuscript now includes detailed annotations and citations for each metric. For more information, please refer to "2.4 Methods" and "References" in the manuscript.

2. Inconsistency between Table 2's Title and Content: The title of Table 2 is “Top 25 city pairs in terms of intensity of population flow during the Spring Festival and daily period,” but the table only lists 10 pairs of cities for each of the two periods. The title should be revised to accurately reflect the content of the table, or the authors should explain why only 10 pairs are included.

The author’s answer: Thank you for your careful review. We apologize for the previous inaccuracy. After carefully checking the raw data and manuscript content, We have revised the title of Table 2 to “Top 10 city pairs in terms of intensity of population flow during the Spring Festival and daily period.”

3. Unclear Description of “Intentional Perturbation”: In section 3.2.1, the term “intentional perturbation” is mentioned, but its precise meaning and application are unclear. The paper does not specify which cities were targeted by the perturbation or provide any details on how the perturbation was implemented. It would be helpful to clarify the definition of “intentional perturbation” and provide further explanation of its implementation and impact, so that readers can better understand the research design.

The author’s answer: Thank you for your valuable suggestions. We apologize for the lack of specificity in the previous formulation of the disturbance scenario. Based on the research context and field practice, the English expression of “intentional perturbation” is revised to “targeted disruption” to clarify the goal-oriented nature of the disturbance. After the revision, the terminology of the whole article is unified to avoid ambiguity.

In the revised article, “2.4.2 Methods of scenario simulation” is added in the “2.4 Methods” section, which illustrates the simulation logic of normal and disruption scenarios. Please refer to “(2) Disruption scenario: dynamic characteristics analysis of network resilience” for explanations of the conceptual connotation, implementation method, measurement indices and their roles of random and targeted disruptions. The following table is a brief description of the random disruption and targeted disruption in the scenario simulation. Please see part (2) in section 2.4.2 of the paper for further details.

Table. Brief description of random and targeted disruptions

Disruption scenario Academic definition Scientific significance Real-world example

Random disruption Randomly remove network nodes and associated edges with uniform probability to simulate unpredictable sudden shocks. Evaluate the resilience change of networks to random failures. Random interruption of road traffic caused by an earthquake.

Targeted disruption Remove nodes and associated edges in descending order of importance to simulate intentional attacks. Identify core network nodes and cascading failure risks. Precision strikes on hub cities during wartime.

Overall, the manuscript presents a valuable analysis framework, but additional details and clearer references are needed to enhance the scientific rigor of the paper.

The author’s answer: We extend our heartfelt gratitude for your thorough review and insightful comments, which have significantly improved the study. We have addressed each of your suggestions and revised the manuscript accordingly. Your emphasis on scientific rigor has deepened our appreciation for precision in research, and we will carry these lessons forward as valuable guidance for our future scholarly endeavors.

Response to Reviewer #2

Many studies have used population flow data to construct urban networks and analyze their structure and resilience characteristics. This article is not innovative enough in terms of methodological design, selection of network structure indicators and resilience measurement, and it does not draw interesting or outstanding results. In addition, I don't see “The static characteristics of network resilience in natural scenarios were analyzed” mentioned by the author in the abstract. The authors just analyzed the structural characteristics of networks using some common network metrics. Specifically, the article has the following deficiencies:

1. There are some problems in the structural arrangement of the article. Usually, the authors should first further discuss the main findings of the study in the discussion section, such as comparing their own research methods, contents and conclusions with the current study, so as to better reflect the innovation of the article. Then in the final section of conclusion, the authors should summarize the methodology, conclusions, research significance, policy implications, and outlook for future research.

The author’s answer: We greatly appreciate your insightful comments and suggestions. The contents of sections "4 Discussion" and "5 Conclusion" have been thoroughly revised according to the comments. Specifically, this includes: (1) Section "4.1 Research significance and characteristics of population flow network resilience" clarifies the main findings of this study. It also compares the research methodology, content, and conclusions with the current research, and identifies the similarities and differences, as well as the reasons for them. The theoretical significance of this study in the design of the theoretical framework and the positive impact on reality are explained. (2) Referring to the common structure of the papers published in PLOS One, the manuscript also writes "4.2 Policy implications" and "4.3 Limitations and future directions" in the discussion. (3) In "5 Conclusion", the research methodology and significance of this study will be condensed, and the conclusion will be summarized more comprehensively.

Reasons for choosing population mobility data for this study and possible innovations: (1) Population, as a key carrier of all kinds of mobility factors such as economy, transportation and culture, has always been an important basis for studying the dynamic development of cities and society. At present, despite the slowing trend of global population growth, the increase of mobile population in some regions continues to rise. As a globally important economic growth pole, Jiangsu, Zhejiang and Shanghai have long ranked among the top regions in China in terms of mobile population size, which makes the region an ideal sample for studying the structural characteristics and resilience response mechanisms of urban networks. Therefore, although there are existing studies that use population mobility data to conduct relevant analyses, the perspective of this study is still of unique value. (2) This study fully absorbs and integrates the existing research methods, constructs the research framework of "structure measurement-scenario simulation-resilience assessment", and systematically reveals the resilience response mechanism of urban networks through static and dynamic scenario simulation. This provides a meaningful research paradigm for urban network resilience research. (3) Based on the urban network resilience assessment results, a "resilience-oriented" regional policy design path is proposed, which provides an empirical basis for breaking the administrative boundary barriers and has strong practical significance. In conclusion, this study not only has innovative perspectives in theory, but also provides important insights for regional coordinated development in practice.

The interesting and notable findings of this study compared to existing research may include: (1) The daily population flow intensity between developed cities is much higher than during the Spring Festival. For example, the average daily mobility intensity between Shanghai and Suzhou on weekdays is 33% higher than during the Spring Festival. This challenges the traditional understanding in classical population mobility models that "festival homecoming tides dominate population mobility intensity", and suggests that the driving force behind population mobility in the Jiangsu-Zhejiang-Shanghai region of China has shifted from "periodic festivals" to "normalized market demands". (2) Results from disruption scenario simulations show that networks based on core nodes are efficient but have high vulnerability. This provides a quantitative tool for urban risk prevention and enriches the analysis of disaster chain networks. (3) Administrative boundaries have a stronger inhibiting effect on links between small and medium-sized cities than between core cities, extending existing discussions on the impact of administrative divisions. It also proposes a "resilience-oriented" regional policy design pathway that provides empirical evidence for breaking down administrative boundary barriers. (4) Detailed discussions and summaries of the above findings are presented in "Section 4 Discussion" and "Section 5 Conclusions".

The reviewer's comment "The static characteristics of network resilience in natural scenarios were analyzed" mainly refers to the third research result in the abstract. To improve clarity and rigor, we have optimized the language in the Abstract section. In addition, "analyzing the static characteristics of network resilience in natural scenarios" involves selecting four indicators - centrality, hierarchy, agglomeration, and transportability - by reviewing, comparing, and summarizing core metrics from the classical literature to analyze network characteristics. Guided by the research framework, this analysis systematically examines the importance of nodes in the network, the hierarchical and redundant nature of the division of labor system, the tightness of connections within subgroups, and the flow efficiency of network elements. Thus, we study the static characteristics of the network under normal conditions-i.e., the functional structural features and operational patterns of the urban network provide clues for optimizing regional development layouts. Specific conceptual definitions and research methods can be found in "(1) Normal scenario: static characteristics analysis of network resilience" within "2.4.2 Methods of scenario simulation", and the research results are detailed in "3.2.1 Analysis of network structure resilience under normal scenario".

2. There are some formatting problems in the reference.

The author’s answer: Thank you for your valuable suggestions. According to the Submission Guidelines of PLOS One, we used the latest style file "PLoS.ens" provided by EndNote to format the references after carefully reviewing the content of the article.

3. There are some language specification problems, for example, in line 171, the terms “betweenness centrality’ and “closeness centrality” seem to be more recognized than “mediating centrality” and “proximity centrality” used in this article.

The author’s answer: Thank you very much for your valuable comments. According to your suggestions and the authoritative references, the relevant terms in the manuscript have been expressed as “betweenness centrality” and “closeness centrality” by the norms of academic terminology. We have also double-checked and revised the expression of other specialized terms in the manuscript.

4. There are inconsistencies in the use of the same term, e.g., whether “toughness” or “resilience’ is used for resilience?

The author’s answer: Your constructive feedback is greatly appreciated. Based on the study’s core framework and authoritative academic literature, the manuscript has standardized the expression of relevant concepts under academic terminology norms. Specifically, this study uniformly uses "resilience", which accurately summarizes the resilience, adaptive capacity, and transformational potential of urban networks from the perspective of dynamic systems theory. In contrast, “resilience” is more in line with the theoretical core and analytical dimensions of this study than “resilience”.

5. In the part 3.2.1, line 353: What is “compose centrality”? What is the basis for normalizing the three centrality indicators to obtain the so-called “compatible centrality”? The authors should provide some supporting evidence, such as references. Also, in the caption of Figure 5, “compose centrality” is written as “integrated centrality”.

The author’s answer: We sincerely appreciate your valuable suggestions. There may have been differences in linguistic translation in the previous expression. Based on the core meaning of this indicator and after consulting authoritative references, we have standardized the terminology as "composite centrality" and cited the relevant literature in the text. For details, please refer to "2.4.3 Methods of measuring the network structural resilience" in the article, where the last paragraph of "(1) Centrality" elaborates on the connotation, role, and calculation method of "composite centrality."

Conceptual connotation: Composite centrality comprehensively reflects a node’s breadth of connections, hub control capacity, and information transmission efficiency. This study comprehensively applies degree centrality, betweenness centrality, and closeness centrality to construct the "composite centrality", providing a multi-dimensional perspective to fully characterize the role of cities. It helps to avoid evaluation biases caused by single-dimensional centrality indicators. For example, it can meticulously identify "high-impact but low-connectivity" potentially vulnerable nodes.

Calculation method: This study first uses z-score standardization to eliminate dimensional differences among the three indicators (degree centrality, betweenness centrality, and closeness centrality). It then applies the entropy weight method to calculate their weight coefficients, thereby deriving the composite centrality for each node. Correspondingly, we have revised the legend of Figure 5 to standardize the terminology for "betweenness centrality," "closeness centrality," and "composite centrality." We have also carefully reviewed the content of all other figures in the article.

Response to Reviewer #3

The paper explores the spatiotemporal dynamics of population flow and network resilience within the Jiangsu-Zhejiang-Shanghai region using one-year Baidu daily migration scale index datasets from 2023. The topic is very interesting. There are some issues that need to be addressed.

1. The data used in the manuscript is th

---

## [Decision Letter · Decision Letter 1]

Jun 26 2025

Dear Dr. Li,

Thank you for submitting your manuscript to PLOS ONE. After careful consideration, we feel that it has merit but does not fully meet PLOS ONE’s publication criteria as it currently stands. Therefore, we invite you to submit a revised version of the manuscript that addresses the points raised during the review process.

We look forward to receiving your revised manuscript.

Kind regards,

Qiwei Ma

Academic Editor

PLOS ONE

Journal Requirements:

Reviewers' comments:

Reviewer's Responses to Questions

**Comments to the Author**

Reviewer #1: All comments have been addressed

Reviewer #2: All comments have been addressed

Reviewer #3: All comments have been addressed

2. Is the manuscript technically sound, and do the data support the conclusions?

Reviewer #1: Yes

Reviewer #2: Yes

Reviewer #3: (No Response)

3. Has the statistical analysis been performed appropriately and rigorously?

Reviewer #1: Yes

Reviewer #2: Yes

Reviewer #3: (No Response)

4. Have the authors made all data underlying the findings in their manuscript fully available?

Reviewer #1: Yes

Reviewer #2: Yes

Reviewer #3: (No Response)

5. Is the manuscript presented in an intelligible fashion and written in standard English?

Reviewer #1: Yes

Reviewer #2: Yes

Reviewer #3: (No Response)

---

## [Author Response · Author response to Decision Letter 2]

19 May 2025

Response Letter

Dear Editors and Reviewers:

We sincerely appreciate your insightful comments and professional suggestions on our manuscript entitled “Exploring the Spatiotemporal Dynamics and Resilience Assessment of Urban Networks from the Perspective of Population Flow” (Manuscript Number: PONE-D-25-01272). Your invaluable feedback has not only significantly improved the quality of this manuscript but also provided important guidance for our ongoing research. In response to your comments, we have undertaken comprehensive revisions to the manuscript, which include but are not limited to:

1. Standardized section numbering and verified figure accuracy.

2. Refined Sections 4.2 and 4.3 to enhance clarity and rigor.

3. Reviewed and updated “References” to comply with journal guidelines.

The following section provides a point-by-point response to each of your specific comments. We believe these revisions have strengthened the manuscript substantially, and we are grateful for your time and expertise. Please do not hesitate to contact us if you require any additional information or clarification.

Yours sincerely,

Jiulin Li Wenhui Lin Jinlong Chu

Response to the comments

Thank you for your thorough revisions and for addressing the reviewer's comments in detail. Upon reviewing the revised manuscript, I appreciate the many positive and effective improvements you have made, which have significantly enhanced the quality of the paper. I especially commend your efforts in clarifying the methodological details and terminology, which have further strengthened the scientific rigor and readability of the manuscript.

However, after another review, I noticed the following points that need further attention and clarification:

1. In Figures 3, 4, and 7, Taizou (JS) is positioned at the bottom, and Taizhou (ZJ) is at the top. However, in Figures 1 and 5, Taizhou (JS) is at the top, and Taizhou (ZJ) is at the bottom. This inconsistency in figure layout should be addressed. I recommend ensuring uniformity in the positioning of the cities across all figures.

The author’s answer: Thank you for bringing this important issue to our attention. We apologize for the inconsistency in the figure layout. We have re-examined the position labels of all the figures to ensure consistency in the geographical location of the cities. The layout of Figures 3, 4, and 7 has been revised to show Taizhou (JS) at the top and Taizhou (ZJ) at the bottom, consistent with Figures 1 and 5. We paid special attention to the standardization and consistency of the study, and also carefully checked the content of the other figures in the manuscript. This revision ensures that all spatial information is conveyed accurately. We hope that the information covered in the study has been fully communicated to you.

2. I noticed that two sections are titled “3.2.1.” However, there is no section 3.2.2 in Chapter 3.2. Please check if the numbering should be revised or if a section is missing, and adjust the titles accordingly.

The author’s answer: Thank you for your valuable comments. The section numbering issue you pointed out is important. Upon reviewing the section structure of the entire text, we discovered a duplication of numbering in section 3.2 of the original draft. Therefore, we have changed the second subsection from "3.2.1" to "3.2.2". We have reorganized the numbering system of section 3 to ensure that the subsequent levels correspond to each other. These revisions have clarified and rationalized the chapter structure of the thesis. Thank you for your help in improving the organization and readability of the thesis.

3. It is unclear whether the resilience calculation is based on daily population flow data or on Spring Festival population flow data. Since you emphasized the importance of both states in the static analysis, and considering that both may face different disruptions, it would be beneficial to discuss resilience in the context of both population flow states.

The author’s answer: Thank you very much for your valuable suggestions, which are an important inspiration for the study. Regarding the use of annual daily average population flow data as the basis for resilience calculation, we make the following clarifications:

(1) Data selection and processing:

Urban network resilience is a structural characteristic formed by the long-term evolution of a region. The attributes it reflects, such as the hierarchy of network division of labor, the closeness of subgroup connections, and the effectiveness of factor flow, are the results of mutual interactions of the urban network system in the process of long-term development. Therefore, the selection of annual daily average data (covering 365 days) can effectively avoid the interference of short-term fluctuations in specific periods, such as the Spring Festival (14 days) and the daily period (30 days in November, for example), on the analysis results, so as to reflect the level of urban network resilience more objectively.

(2) Specific analysis process:

①This study analyzes population flow throughout the year, the Spring Festival, and the daily period. It unifies the data comparison benchmarks in different periods by standardizing daily averages. This allows for a deeper exploration of the spatio-temporal characteristics of the urban network. ②Using annual daily average population flow data as the core basis, we reveal the static characteristics of network resilience in normal scenarios and disruption scenarios through quantitative analysis. ③Based on the spatio-temporal patterns and resilience characteristics of the population mobility network, we propose policy recommendations.

(3) Expansion and reflection of research direction:

Your suggestion to explore differences in resilience characteristics in different periods is meaningful and will be an important direction for future research. Due to the limited length of the article, it is difficult for this study to break down periods further to explore resilience in depth. In follow-up research, we plan to develop a dynamic resilience assessment framework based on long time series, which will focus on analyzing resilience performance in the context of extreme flows, such as special periods and events. The revised manuscript of the current study provides additional explanation in section "4.3 Limitations and future directions".

Thank you again for your valuable suggestions, which will greatly help us improve the current study and guide subsequent research.

4. In Section 4.2.2, you suggest that "Lianyungang and Yancheng should focus on upgrading tourism infrastructure and service ecosystems to promote livability-driven growth," among other recommendations. However, these insights do not seem to be directly derived from the analysis of population flow. Could you please provide a clearer explanation of how these industry recommendations are linked to the population flow analysis?

The author’s answer: Thank you very much for your valuable comments. We are keenly aware of the lack of logical rigor in our previous policy recommendations. Regarding the connection between the recommendations and the population mobility network analysis, we offer the following clarifications:

(1) The basis of the policy recommendations:

①Population flow is usually a rational choice based on multidimensional considerations, such as the level of urban development, industrial economic benefits, supporting facilities and services, and regional culture. This study constructs a city network based on population flow and analyzes its spatial-temporal characteristics and resilience level to accurately identify city nodes with weak population attraction, a lack of development momentum, and lower resilience. Combining the results of the study, the current state of urban development and relevant planning documents, we propose suggestions to improve the urban network system in the study area. This will enhance the region's overall development quality and risk-resistant capacity. ②Industry is one of the key factors affecting population mobility and urban development. The results of the network resilience assessment show that northern Jiangsu and southern Zhejiang (including Lianyungang, Suqian, and Quzhou, among other cities) have obvious obstacles in the "core-edge" connection. This results in the region's comprehensive competitiveness being much lower than that of central, developed areas. Despite their unique location advantages and rich ecological and cultural resources, these regions have not realized synergies with developed regions due to their lack of related industries. Therefore, this study considers industry as an important perspective in the policy recommendations.

(2) Revised manuscript description:

①We have systematically optimized the logic and presentation of Section 4.2.2. We have strengthened the linkage with network resilience, and put forward policy recommendations that are more in line with the findings of the study and the reality of the city. Specifically, we added the direct correspondence between network resilience and population mobility recommendations. We emphasize the significance of developing cultural and tourism industries based on location and resource endowment, and upgrading supporting services, for small and medium-sized cities to attract population flows and promote the dynamic equilibrium of regional mobility networks. ②We have also optimized the elaboration of other policy recommendations in Section 4.2, aiming to convey the core ideas of the study more clearly and accurately.

Thank you again for your help in identifying the shortcomings of our argument. Your feedback is crucial to enhancing the rigor and academic value of this study.

Thank you once again for your efforts in improving the manuscript. I believe that addressing these issues will further strengthen the rigor and academic value of the paper.

The author’s answer: We sincerely thank you for your careful review and valuable comments on this study. We have responded to the review comments one by one and revised the relevant contents. And we have done a deep reflection on our scientific rigor. These changes will greatly benefit our future scientific research.

---

## [Editor Report · Decision Letter 2]

Exploring the spatiotemporal dynamics and resilience assessment of urban networks from the perspective of population flow

PONE-D-25-01272R2

Dear Dr. Li,

We’re pleased to inform you that your manuscript has been judged scientifically suitable for publication and will be formally accepted for publication once it meets all outstanding technical requirements.

Kind regards,

Qiwei Ma

Academic Editor

PLOS ONE
---

## [Editor Report · Acceptance letter]

PONE-D-25-01272R2

PLOS ONE

Dear Dr. Li,

I'm pleased to inform you that your manuscript has been deemed suitable for publication in PLOS ONE. Congratulations! Your manuscript is now being handed over to our production team.

Kind regards,

on behalf of

Prof. Qiwei Ma

Academic Editor

PLOS ONE